# Fully Neural Network based Model for General Temporal Point Processes

**Takahiro Omi**
The University of Tokyo, RIKEN AIP
takahiro.omi.em@gmail.com

**Naonori Ueda**
NTT Communication Science Laboratories, RIKEN AIP
naonori.ueda.fr@hco.ntt.co.jp

**Kazuyuki Aihara**
The University of Tokyo
aihara@sat.t.u-tokyo.ac.jp

## Abstract

A temporal point process is a mathematical model for a time series of discrete events, which covers various applications. Recently, recurrent neural network (RNN) based models have been developed for point processes and have been found effective. RNN based models usually assume a specific functional form for the time course of the intensity function of a point process (e.g., exponentially decreasing or increasing with the time since the most recent event). However, such an assumption can restrict the expressive power of the model. We herein propose a novel RNN based model in which the time course of the intensity function is represented in a general manner. In our approach, we first model the integral of the intensity function using a feedforward neural network and then obtain the intensity function as its derivative. This approach enables us to both obtain a flexible model of the intensity function and exactly evaluate the log-likelihood function, which contains the integral of the intensity function, without any numerical approximations. Our model achieves competitive or superior performances compared to the previous state-of-the-art methods for both synthetic and real datasets.

## 1 Introduction

The activity of many diverse systems is characterized as a sequence of temporally discrete events. The examples include financial transactions, communication in a social network, and user activity at a web site. In many cases, the occurrences of the event are correlated to each other in a certain manner, and information on future events may be extracted from the information of past events. Therefore, the appropriate modeling of the dependence of the event occurrence on the history of past events is important for understanding the system and predicting future events.

A temporal point process is a useful mathematical tool for modeling the time series of discrete events. In this framework, the dependence on the event history is characterized using a conditional intensity function that maps the history of the past events to the intensity function of the point process. The most common models, such as the Poisson process or the Hawkes process [1, 2, 3], assume a specific parametric form for the conditional intensity function. Recently, Du *et al.* (2016) proposed a model based on a recurrent neural network (RNN) for point processes [4], and the variant models were further developed [5, 6, 7, 8, 9, 10]. In this approach, an RNN is used to obtain a compact representation of the event history. The conditional intensity function is then modeled as a function of the hidden state of the RNN. Consequently, the RNN based models outperform the parametric models in prediction performance.

Although such RNN based models aim to capture the dependence of the event occurrence on the event history in a general manner, a specific functional form is usually assumed for the time course of the conditional intensity function (see [6, 7] for exception). For example, the model in [4] assumed that the conditional intensity function exponentially decreases or increases with the elapsed time from the most recent event until the next event. However, using such an assumption can limit the expressive ability of the model and potentially deteriorate the predictive skill if the employed assumption is incorrect. We herein generalize RNN based models such that the time evolution of the conditional intensity function is represented in a general manner. For this purpose, we formulate the conditional intensity function based on a neural network rather than assuming a specific functional form.

However, exactly evaluating the log-likelihood function for such a general model is generally intractable because the log-likelihood function of a temporal point process contains the integral of the conditional intensity function. Although some studies, which considered a general model of the intensity function, used numerical approximations to evaluate the integral [6, 7], numerical approximations can deteriorate the fitting accuracy and can also be computationally expensive. To overcome this limitation, we first model the integral of the conditional intensity function using a feedforward neural network rather than directly modeling the conditional intensity function itself. Then, the conditional intensity function is obtained by differentiating it. This approach enables us to exactly evaluate the log-likelihood function of our general model without numerical approximations. Finally, we show the effectiveness of our proposed model by analyzing synthetic and real datasets.

## 2 Method

### 2.1 Temporal point process

A temporal point process is a stochastic process that generates a sequence of discrete events at times $\{t_i\}_{i=1}^n$ in a given observation interval $[0, T]$. The process is characterized via a conditional intensity function $\lambda(t|H_t)$, which is the intensity function of the event at the time $t$ conditioned on the event history $H_t = \{t_i|t_i < t\}$ up to the time $t$, given as follows:

$$\lambda(t|H_t) = \lim_{\Delta \to 0} \frac{P(\text{one event occurs in } [t, t+\Delta]|H_t)}{\Delta}. \tag{1}$$

If the conditional intensity function is specified, the probability density function of the time $t_{i+1}$ of the next event, given the times $\{t_1, t_2, \ldots, t_i\}$ of the past events, is obtained as follows:

$$p(t_{i+1}|t_1, t_2, \ldots, t_i) = \lambda(t_{i+1}|H_{t_{i+1}}) \exp\left\{-\int_{t_i}^{t_{i+1}} \lambda(t|H_t)dt\right\}, \tag{2}$$

where the exponential term in the right-hand side represents the probability that no events occur in $[t_i, t_{i+1}]$. The probability density function to observe an event sequence $\{t_i\}_{i=1}^n$ is then obtained as follows:

$$p(\{t_i\}_{i=1}^n) = \prod_{i=1}^n \lambda(t_i|H_{t_i}) \exp\left\{-\int_0^T \lambda(t|H_t)dt\right\}. \tag{3}$$

The most basic example of a temporal point process is a stationary Poisson process, which assumes that the events are independent of each other. The conditional intensity function of the stationary Poisson process is given as $\lambda(t|H_t) = \lambda$. Another popular example is the Hawkes process [1, 2, 3], which is a simple model of a self-exciting point process. The conditional intensity function of the Hawkes process is given as $\lambda(t|H_t) = \mu + \sum_{t_i < t} g(t - t_i)$, where $g(s)$ is a kernel function ($g(s) = 0$ if $s < 0$) that represents the triggering effect from the past event.

### 2.2 Recurrent neural network approach to a temporal point process

The conditional intensity function, which maps the event history to the intensity function, plays a major role in the modeling of point processes. Du *et al.* (2016) proposed to use an RNN to model the conditional intensity function [4]. In this approach, an input vector $x_i$, which extracts the information of the event time $t_i$, is first fed into the RNN. A simple form of the input is the

inter-event interval as $\boldsymbol{x}_i = (t_i - t_{i-1})$ or its logarithm as $\boldsymbol{x}_i = (\log(t_i - t_{i-1}))$. A hidden state $\boldsymbol{h}_i$ of the RNN is updated as follows:

$$\boldsymbol{h}_i = f(W^h \boldsymbol{h}_{i-1} + W^x \boldsymbol{x}_i + \boldsymbol{b}^h), \tag{4}$$

where $W^h$, $W^x$, and $\boldsymbol{b}^h$ denote the recurrent weight matrix, input weight matrix, and bias term, respectively, and $f$ is an activation function. We here treat the hidden state of the RNN as a compact vector representation of the event history. The conditional intensity function is then formulated as a function of the elapsed time from the most recent event and the hidden state of the RNN, given as follows:

$$\lambda(t|H_t) = \phi(t - t_i | \boldsymbol{h}_i), \tag{5}$$

where $\phi$ is a non-negative function referred to as a *hazard function*.

Du *et al.* (2016) assumed the following form for the hazard function [4] :

$$\phi(\tau | \boldsymbol{h}_i) = \exp(w^t \tau + \boldsymbol{v}^\phi \cdot \boldsymbol{h}_i + b^\phi). \tag{6}$$

The exponential function in the above equation is used to ensure the non-negativity of the intensity. In this model, the conditional intensity function exponentially decreases or increases with the elapsed time $\tau$ from the most recent event until the next event.

A simplified model, where the conditional intensity function is constant over the period between the successive events, was also considered in [8, 10]. We here formulate such a model as a special case of the model of eq. (6), given as follows:

$$\phi(\tau | \boldsymbol{h}_i) = \exp(\boldsymbol{v}^\phi \cdot \boldsymbol{h}_i + b^\phi). \tag{7}$$

In this model, the inter-event interval $\tau_i = t_{i+1} - t_i$ follows the exponential distribution with mean $1 / \exp(\boldsymbol{v}^\phi \cdot \boldsymbol{h}_i + b^\phi)$.

The log-likelihood function of the RNN based model can be obtained from eq. (3) as follows:

$$\log L(\{t_i\}) = \sum_i \left[ \log \phi(t_{i+1} - t_i | \boldsymbol{h}_i) - \int_0^{t_{i+1} - t_i} \phi(\tau | \boldsymbol{h}_i) d\tau \right]. \tag{8}$$

The parameter values of the model are estimated by maximizing the log-likelihood function. For this purpose, the backpropagation through time (BPTT) is employed to obtain the gradient of the log-likelihood function. In the BPTT, the RNN is unfolded to a feedforward network, whose weights are shared across layers, and the backpropagation is applied to the unfolded network. Although the hidden state $\boldsymbol{h}_i$ of the RNN originally depends on all the preceding events, fully considering the history dependence for a long sequence can be problematic due to gradient vanishing or explosion. Only the dependence on a fixed number $d$ of the most recent events is considered herein as was done in [4], where $d$ is a hyperparameter called the truncation depth. Namely, for each time index $i$, the hidden state $\boldsymbol{h}_i$ is obtained by feeding the inputs $\{\boldsymbol{x}_j\}_{j=i-d+1}^i$ from the $d$ most recent events to the RNN.

### 2.3 Problem

The main problem of the previous studies is that a specific functional form is usually assumed for the time course of the hazard function $\phi(\tau | \boldsymbol{h}_i)$ as in eqs. (6) or (7), which can miss the general dependence of the event occurrence on the past events. One may want to exploit a more complex model for the hazard function $\phi(\tau | \boldsymbol{h}_i)$ to generalize the model. However, such a complex model is generally intractable because the log-likelihood function in eq. (8) includes the integral of the hazard function. Although the integral may be approximately evaluated using numerical methods [6, 7], the numerical approximations can deteriorate the fitting accuracy and be computationally expensive. This is the main limitation in the flexible modeling of the hazard function.

### 2.4 The proposed model [1]

Rather than directly modeling the hazard function, we herein propose to model the *cumulative hazard function* $\Phi(\tau | \boldsymbol{h}_i)$, defined as follows:

$$\Phi(\tau | \boldsymbol{h}_i) = \int_0^\tau \phi(s | \boldsymbol{h}_i) ds. \tag{9}$$

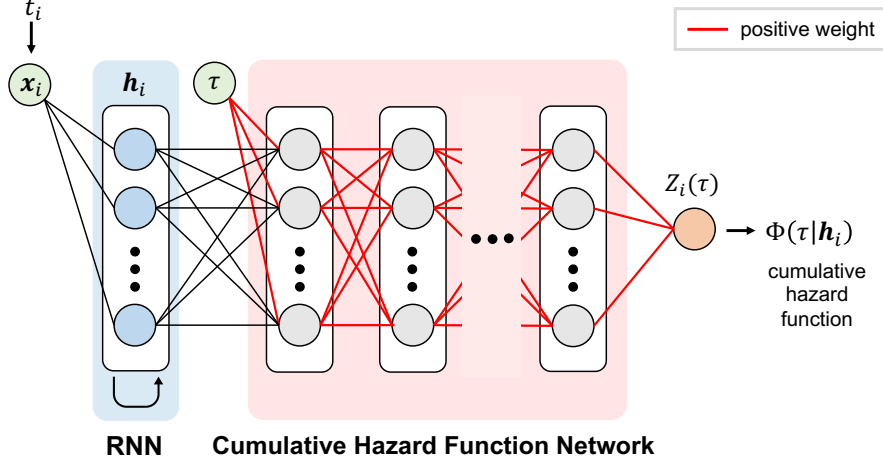

Figure 1: Network architecture to output the cumulative hazard function.

The hazard function itself can be then obtained by differentiating the cumulative hazard function with respect to $\tau$ as follows:

$$\phi(\tau|\boldsymbol{h}_i) = \frac{\partial}{\partial \tau}\Phi(\tau|\boldsymbol{h}_i). \tag{10}$$

The log-likelihood function is reformulated as follows using the cumulative hazard function:

$$\log L(\{t_i\}) = \sum_i \left[ \log\left\{ \frac{\partial}{\partial \tau}\Phi(\tau = t_{i+1} - t_i|\boldsymbol{h}_i) \right\} - \Phi(t_{i+1} - t_i|\boldsymbol{h}_i) \right]. \tag{11}$$

Now, the log-likelihood function does not include the integral term in contrast to eq. (8) and can be exactly evaluated even for a complex model of the cumulative hazard function.

In the present study, we model the cumulative hazard function using a feedforward neural network (*a cumulative hazard function network*; Fig. 1) for flexible modeling. The cumulative hazard function is a monotonically increasing function of $\tau$ and is positive-valued. The cumulative hazard function network is designed to reproduce these properties. The positivity of the network output can be ensured using an output unit, in which the activation function is positive-valued. Considering monotonicity, we employ the idea used in [11, 12]. To summarize, the weights of the particular network connections are constrained to be positive (Fig. 1).

The detail of the cumulative hazard function network is described below. In the network, each unit receives the weighted sum of the inputs and applies an activation function to produce the output. The first hidden layer in the network receives the elapsed time $\tau$ and the hidden state $\boldsymbol{h}_i$ of the RNN as the inputs[2]. The weights of the connections from the elapsed time $\tau$ to the first hidden layer and all the connections from the hidden layers are constrained to be positive[3]. The connections, in which the weights are constrained to be positive, are indicated by the red line in Fig. 1. The activation functions of the hidden units and the output unit are set to be the $tanh$ function and the $softplus$ function, $\log(1 + \exp(\cdot))$, respectively. In this setting, the network output is monotonically increasing with respect to the elapsed time $\tau$ and takes only a positive value, which mimics the cumulative hazard function.

The cumulative hazard function $\Phi(\tau|\boldsymbol{h}_i)$ and the hazard function $\phi(\tau|\boldsymbol{h}_i)$ are now formulated as follows based on the output $Z_i(\tau)$ of the cumulative hazard function network:

$$\Phi(\tau|\boldsymbol{h}_i) = Z_i(\tau), \tag{12}$$

$$\phi(\tau|\boldsymbol{h}_i) = \frac{\partial}{\partial \tau}\Phi(\tau|\boldsymbol{h}_i) = \frac{\partial}{\partial \tau}Z_i(\tau). \tag{13}$$

The differentiation term $\partial Z_i(\tau)/\partial \tau$, which is the derivative of the network output with respect to the network input, is computed using automatic differentiation [13, 14] (see Supplementary Material for more details). Automatic differentiation is a method to calculate the derivative of an arbitrary function, and it can be easily carried out using neural network libraries such as TensorFlow and PyTorch. The log-likelihood function in eq. (11) of our model is then given based on the network output via the eqs. (12) and (13). The gradient of the log-likelihood function with respect to the parameters is obtained using backpropagation (see Supplementary Material for more details).

We note that our model can also efficiently generate a prediction of the timing of the coming event in the following way. The predictive probability density function $p^*(t|t_1, t_2, \ldots, t_i)$ of the time $t_{i+1}$ of the coming event given the past events $\{t_1, t_2, \ldots, t_i\}$ is calculated from eq. (2). We herein use the median $t^*_{i+1}$ of the predictive distribution $p^*$ to predict $t_{i+1}$. To obtain the median $t^*_{i+1}$, we use the relation $\Phi(t^*_{i+1} - t_i|\boldsymbol{h}_i) = \log(2)$. This relation is derived from the property that the integral of the intensity function over $[t_i, t_{i+1}]$ follows the exponential distribution with mean 1, or is derived by directly integrating eq. (2). Then, the median $t^*_{i+1}$ can be efficiently obtained by solving the above relation using a root finding method (e.g., the bisection method); it takes only a second for our model to generate predictions for 20000 events. Therefore, the cumulative hazard function also plays a crucial role in generating a median predictor.

## 3 Related works

The RNN based point process models were proposed in [4]. Most previous studies assumed a specific functional form for the time-course of the hazard function. The exponential hazard function in eq. (6) is commonly assumed [4, 9]. Some studies [8, 10] assumed the constant hazard function as in eq. (7), which is equivalent to assume that the inter-event intervals follow the exponential distribution. In contrast to these studies, our model does not assume any specific functional form for the hazard function, and the time course of the hazard function is formulated in a general manner based on a neural network.

A few studies addressed the general modeling of the hazard function. Jing and Somla (2017) proposed to discretize the continuous hazard function to a piecewise constant function [6], given as follows:

$$\phi(\tau|\boldsymbol{h}_i) = softplus(\boldsymbol{v}^p_j \cdot \boldsymbol{h}_i + b^p_j) \quad \text{if } (j-1)l < \tau \le jl, \tag{14}$$

for $j = 1, 2, \ldots, \tau_{max}/l$ for some choice of $l$ and $\tau_{max}$. Mei and Eisner (2017) proposed a continuous-time long short-term memory (LSTM) where the output continuously evolves in time, and the conditional intensity function is given as a function of the output [7]. This model used the Monte Carlo method to approximate the integral. In all cases, numerical approximations are used to evaluate the integral in the log-likelihood function in eq. (8). However, numerical approximations can be computationally expensive and can also affect the fitting accuracy. In contrast to these studies, the log-likelihood function of our general model can be exactly evaluated without any numerical approximations because the integral of the hazard function is modeled by a feedforward neural network in our approach. Therefore, a more accurate estimate can be efficiently obtained by our approach.

## 4 Experiments

In this section, we conduct experiments using synthetic and real data. We herein evaluate the predictive performances of the four RNN based point process models. The number of units in the RNN is fixed to 64 for all the models. The first two models are equipped with the constant hazard function in eq. (7) (**the constant model**) and the exponential hazard function in eq. (6) (**the exponential model**), respectively. The third model employs the piecewise constant hazard function in eq. (14) (**the piecewise constant model**). We set $\tau_{max}$ to the maximum value of the inter-event interval in each dataset and use the condition $l = \tau_{max}/128$. The fourth model employs the neural network based hazard function proposed in this study (**the neural network based model**). For this model, we use two hidden layers for the cumulative hazard function network, and the number of units in each layer is 64. In this setting, the numbers of the parameters are almost the same between the third and fourth models.

Each dataset is divided into the training and test data. In the training phase, the model parameters are estimated using the training data. For this optimization, the Adam optimizer with the learning rate 0.001, $\beta_1 = 0.9$, and $\beta_2 = 0.999$ is used [16], and the batch size is 256. We also choose the truncation depth $d$ from $[5, 10, 20, 40]$ using 20% of the training data (see Sec. 2.2 for more details on truncation depth). In the test phase, we evaluate the predictive performances of the trained models using the test data. In each time step, the probability density function $p^*(t|t_1, t_2, \ldots, t_i)$ of the time of the coming event given the past events $\{t_1, t_2, \ldots, t_i\}$ is calculated from eq. (2), and scored by the negative log-likelihood $-\log p^*(t_{i+1}|t_1, t_2, \ldots, t_i)$ for the actually observed $t_{i+1}$ (a smaller score means a better predictive performance). The score is finally averaged over the events in the test data. We performed these computations under a GPU environment provided by Google Colaboratory.

## 4.1 Synthetic data

We use the synthetic data generated from the following stochastic processes. In this experiment, 100,000 events are generated from each process, and 80,000/20,000 events are used for training/testing.

**Stationary Poisson Process (S-Poisson):** The conditional intensity function is given as $\lambda(t|H_t) = 1$.

**Non-stationary Poisson Process (N-Poisson):** The conditional intensity function is given as $\lambda(t|H_t) = 0.99 \sin(2\pi t/20000) + 1$.

**Stationary Renewal Process (S-Renewal):** In this process, the inter-event intervals $\{\tau_i = t_{i+1} - t_i\}$ are independent and identically distributed according to a given probability density function $p(\tau)$. We herein use the log-normal distribution with a mean of 1.0 and a standard deviation of 6.0 for $p(\tau)$. In this setting, a generated sequence exhibits a bursty behavior: multiple events tend to occur in a short period and are followed by a long silence period like the burst firing of biological neurons.

**Non-stationary Renewal Process (N-Renewal):** A sequence $\{t_i\}$ following a non-stationary renewal process is obtained as follows [17]: we first generate a sequence $\{t'_i\}$ from a stationary renewal process, and then we rescale the time according to $t'_i = \int_0^{t_i} r(t)dt$ for a non-negative trend function $r(t)$. We use the gamma distribution with a mean of 1.0 and a standard deviation of 0.5 to generate the stationary renewal process and set the trend function to $r(t) = 0.99 \sin(2\pi t/20000) + 1$. In this process, an inter-event interval tends to be followed by the one with similar length, but the expected length gradually varies in time.

**Self-correcting Process (SC):** The conditional intensity function is given as $\lambda(t|H_t) = \exp(t - \sum_{t_i < t} 1)$.

**Hawkes Processes (Hawkes1 and Hawkes2):** We use the Hawkes process, in which the kernel function is given by the sum of multiple exponential functions: the conditional intensity function is given by $\lambda(t|H_t) = \mu + \sum_{t_i < t} \sum_{j=1}^{M} \alpha_j \beta_j \exp\{-\beta_j(t - t_i)\}$. For the **Hawkes1** model, we set $M = 1, \mu = 0.2, \alpha_1 = 0.8$, and $\beta_1 = 1.0$. For the **Hawkes2** model, we set $M = 2, \mu = 0.2, \alpha_1 = 0.4, \beta_1 = 1.0, \alpha_2 = 0.4$, and $\beta_2 = 20.0$. Compared to the Hawkes1 model, the kernel function of the Hawkes2 model rapidly varies in time for small $\tau$.

In addition to the four RNN based models, we evaluate the predictive performance of the true model, i.e., the model that generated the data, and use it as a reference. The scores of the RNN based models are standardized by subtracting the score of the true model. The value of 0 in the standardized score corresponds to the score of the true model.

Figure 2 summarizes the performances of the four RNN based models for the synthetic datasets (a smaller score means a better performance). We first find that the proposed neural network based model achieves a competitive or better performance against performances of the other models. The neural network based model also performs robustly for all the datasets: the performance of the neural network based model is always close to that of the true model. These results demonstrate that (i) the performance is improved by employing the neural network based hazard function and that (ii) our model can be applicable to a diverse class of data generating processes.

The performances of the constant model and the exponential model critically depend on whether the hazard function is correctly specified. The constant hazard function is correct for the S-Poisson and

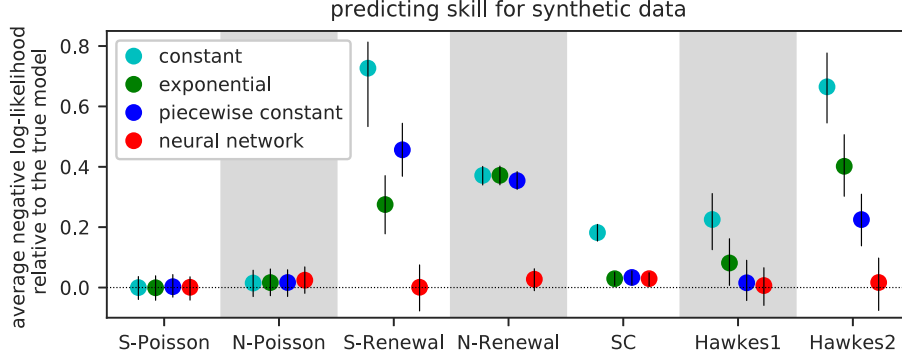

Figure 2: Performances for the synthetic datasets. For each synthetic dataset, we evaluate the predictive performances of the four RNN based models according to the negative log-likelihood averaged per event calculated for the test data (the smaller is more preferable). Each score in the panel is standardized by subtracting the score of the true model. Each error bar represents the 25% and 75% percentiles of the score calculated for every 300 samples.

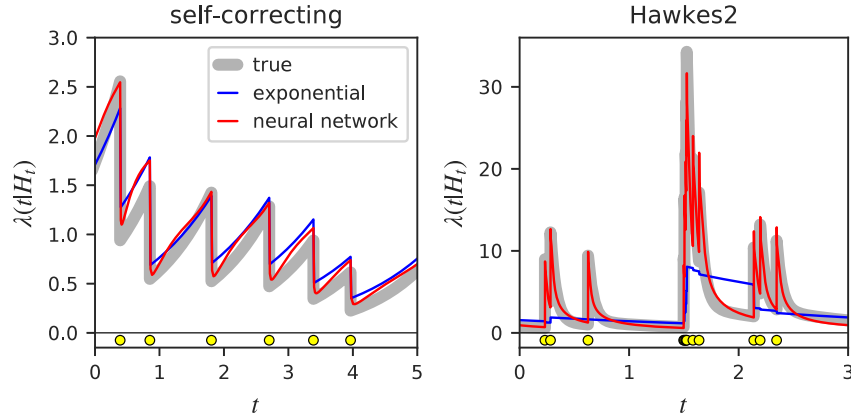

Figure 3: The conditional intensity functions estimated from the exponential model and the neural network based model are compared with the true one. The yellow circles in the bottom of each panel represent the events. The exponential hazard function is valid for the self-correcting process, but not for the Hawkes2 process.

N-Poisson processes. The exponential hazard function is correct for the S-Poisson, N-Poisson, and SC processes, and is approximately correct for the Hawkes1 process (a constant term is included in the hazard function of the Hawkes1 process but not in the exponential hazard function). In fact, these models perform similarly to the true model for the cases where the hazard function is correctly specified but perform poorly for the other cases. Figure 3 shows the estimated conditional intensity function and clearly demonstrates that the exponential model captures well the true conditional intensity function for the self-correcting process where the exponential hazard function is valid; however, it fails for the Hawkes2 process where the exponential hazard function is not valid. In contrast, our neural network based model can reproduce well the true model for both cases. In this manner, the constant and exponential models are sensitive to model misspecification.

The performance of the piecewise constant model is much worse than the neural network based model, particularly for the S-Renewal, N-Renewal, and Hawkes2 processes. For these processes, the variability of the inter-event intervals is large, and the conditional intensity function can rapidly vary for a short period after an event. For such cases, the piecewise constant approximation might not work well. The performance of the piecewise constant model would be improved if the approximation accuracy is improved; however, this increases the computational cost. In this experiment, the numbers of the parameters are set to be almost the same between the piecewise constant model and the neural network based model. Moreover, the neural network based model performs better

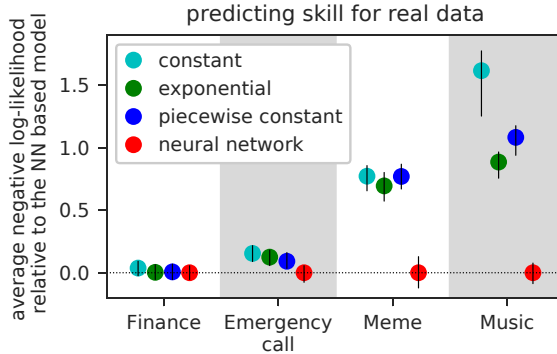

Figure 4: Performances for the real datasets. The score of each RNN based model is standardized by subtracting the score of the neural network based model.

than the piecewise constant model, indicating that the neural network based model is more efficient than the piecewise constant model.

## 4.2 Real data

We use the following real datasets for the next experiment.

**Finance dataset:** This dataset contains the trading records of Nikkei 225 mini, which is the most liquid features contracts in Asia [18]. The timestamps of 182,373 transactions in one day are analyzed, and the first $80\%$ and the last $20\%$ of the data are used for training and testing, respectively.

**Emergency call dataset:** This dataset contains the records of the police department calls for service in San Francisco [19]. Each record contains the timestamp and address from which the call was made. We prepare 100 separate sequences for the 100 most frequent addresses, which contain a total of 294,865 events. The first $80\%$ and the last $20\%$ of the events in the sequences are used for training and testing, respectively.

**Meme dataset:** MemeTracker tracks the popular phrases from numerous online resources such as news media and personal blogs [20]. This dataset records the timestamps when the focused phrases appear on the internet. We first extract the 50 most frequent phrases and obtain the corresponding 50 separate sequences. We use 40 sequences out of 50 with 247,579 events for training and the remaining 10 sequences with 61,095 events for testing.

**Music dataset:** This dataset records the history of music listening of users at `https://www.last.fm/` [21]. We prepare the 100 sequences for the 100 most active users in Jan 2009, which contain a total of 299,046 events. The first $80\%$ and the last $20\%$ of the events in the sequences are used for training and testing, respectively.

Figure 4 summarizes the performances for the real datasets. We find that the neural network based model exhibits a competitive or superior score as compared to those of the other models; this demonstrates the practical effectiveness of the proposed model. For the finance dataset, the performances of all the models are close to each other, implying that the constant or exponential hazard function reproduces the event occurrence process of the financial transactions. The neural network based model performs much better than the other models, particularly for the Meme and music datasets: the difference in the scores between the neural network based model and the other models is greater than 0.5. This difference should be significant (e.g., in the case of the right panel in Fig. 3, where the exponential model clearly fails to reproduce the true model, the score difference is about 0.4 between the true and exponential models). For the two datasets, the data contain inter-event intervals that are much longer than the average, and the variability of the inter-event intervals is large. Other than the neural network based model, the models presumably fail to adapt to such a feature.

## 4.3 Time prediction experiment

To evaluate the predictive performances based on a metric other than the log-likelihood, we also carry out the time prediction experiments. Specifically, we use the median of the predictive distri-

bution to predict the timing of the coming event and evaluate the prediction by the mean absolute error. The result is summarized in the table below. In the table, the best score is in bold, and it is in red if the difference between the best score and the second best score is statistically significant ($p < 0.01$). We find that our model performs better than the other models on average and that the performance of our model is best or close to the best for the most individual datasets. These results demonstrate the effectiveness of our model in the prediction task.

| | S-Poisson | N-Poisson | S-Renewal | N-Renewal | SC | Hakes1 | Hawkes2 | Finance | Call | Meme | Music | AVERAGE |
|---|---|---|---|---|---|---|---|---|---|---|---|---|
| constant | 0.696 | **0.704** | 1.085 | 0.450 | 0.543 | 0.915 | 1.075 | 0.883 | 0.876 | 0.848 | 1.066 | 0.831 |
| exponential | 0.696 | 0.716 | 0.917 | 0.452 | 0.498 | 0.854 | 0.986 | 0.840 | 0.861 | 0.823 | 0.788 | 0.766 |
| piecewise const. | **0.696** | 0.716 | 0.982 | 0.437 | **0.494** | 0.850 | 0.965 | **0.839** | 0.862 | 0.826 | 0.856 | 0.775 |
| neural network | 0.696 | 0.710 | **0.894** | **0.414** | 0.496 | **0.848** | **0.962** | 0.847 | **0.853** | **0.811** | **0.783** | **0.756** |

## 4.4 Comparison with the continuous-time LSTM model

We here compare our model with the continuous-time LSTM (CT-LSTM) model [7]. Both the two models aim at flexibly estimating the intensity function. The technical advantage of our model over the CT-LSTM model is that the log-likelihood function can be exactly evaluated, therefore the estimation can be carried out efficiently. Our model is also relatively easy to implement. For the CT-LSTM model, the evaluation of the log-likelihood function is based on the Monte Carlo method, which can be computationally expensive and can deteriorate the performance.

We evaluate the predictive performance of the CT-LSTM model in terms of the mean negative log-likelihood (MNLL) and the mean absolute error (MAE). The number of the hidden units in the CT-LSTM model is set to 42, so that the numbers of the free parameters are almost the same between the CT-LSTM model and our model. The following table lists the scores of the CT-LSTM model relative to our model; the positive score means that our model is better than the CT-LSTM model. In the table, the score is in blue if our model is significantly better than the CT-LSTM model ($p < 0.01$). The performance of our model is better than the CT-LSTM model on average, demonstrating the effectiveness of our model.

| | S-Poisson | N-Poisson | S-Renewal | N-Renewal | SC | Hakes1 | hawkes2 | Finance | Call | Meme | Music | AVERAGE |
|---|---|---|---|---|---|---|---|---|---|---|---|---|
| MNLL | -0.002 | -0.013 | 0.012 | -0.007 | -0.008 | -0.002 | 0.002 | -0.011 | -0.017 | 0.590 | 0.163 | 0.064 |
| MAE | 0.000 | -0.007 | 0.000 | 0.042 | 0.016 | 0.002 | 0.005 | -0.013 | -0.002 | -0.002 | 0.026 | 0.009 |

# 5 Discussion and Conclusions

In this study, we extended the RNN based point process models such that the time course of the hazard function is represented in a general manner based on a neural network. We then showed the effectiveness of our model by analyzing both synthetic and real datasets. Primary advantages of the proposed model are summarized as follows:

- By using a feedforward neural network, our model can reproduce any time course of the hazard function in principle, i.e., the usefulness of fully neural network based modeling for point processes is indicated.

- By modeling the cumulative hazard function rather than the hazard function itself, we can avoid the direct evaluation of the integral in the log-likelihood function. The log-likelihood function can be exactly and efficiently evaluated without relying on the numerical approximation because of this approach.

We note that the cumulative hazard function also plays an important role in the diagnostic analysis [22, 23, 24]. We did not consider herein the marks of each event (i.e., the information associated with each event other than the timestamp) because the primary contribution of this work is the development of a general model of the hazard function. However, our approach can be easily extended to marked temporal point processes as well.

**Acknowledgments**

This research was partly supported by AMED under Grant Number JP19dm0307009. T. O. and K. A. are supported by Kozo Keikaku Engineering Inc.

## Footnotes

[1] A source code is available online. *https://github.com/omitakahiro/NeuralNetworkPointProcess*

[2]We may input $\log \tau$ to the first layer rather than $\tau$ if the variation of the inter-event interval is large.

[3]In order to enforce the weights to be positive, if a weight is updated to be a negative value during training, we replace it with its absolute value.

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
