[Supplementary Material]

## Supplementary Material for "Fully Neural Network based Model for General Temporal Point Processes"

**Evaluating the $\partial Z_i(\tau)/\partial\tau$**

In our method, we need to calculate $\partial Z_i(\tau)/\partial\tau$, the derivative of the output $Z_i(\tau)$ of the cumulative hazard function network with respect to the input $\tau$. The derivative $\partial Z_i(\tau)/\partial\tau$ can be computed using automatic differentiation. Automatic differentiation is a method to evaluate the derivative of an arbitrary function [13]. We here employ backpropagation which is a special case of automatic differentiation. Backpropagation is sometimes considered as a specific method to evaluate the gradient of the loss function for a multi-layer perceptron, however backpropagation is a fairly general method to compute the gradient of an arbitrary function. Please refer to [14] for more details on backpropagation.

We first formulate the operation of the cumulative hazard function network. We denote the output of the $j$th layer in the cumulative hazard function network by $\boldsymbol{Y}^{(j)} \in \mathbb{R}^{m_j}$ and write the feedforward operation as follows:

$$\boldsymbol{Y}^{(j)} = f^{(j)}(W^{(j)}\boldsymbol{Y}^{(j-1)} + \boldsymbol{b}^{(j)}) \qquad (j = 1, 2, \ldots, L), \tag{15}$$

where $f^{(j)}$, $W^{(j)}$, and $\boldsymbol{b}^{(j)}$ represent the activation function, the weight matrix, and the bias term for the $j$th layer, respectively. The input to the cumulative hazard function network is given as $\boldsymbol{Y}^{(0)} = (\tau, \boldsymbol{h}_i^T)^T$, where $\boldsymbol{h}_i$ is the RNN output. The $L$th layer is the output layer, and we have $Z_i(\tau) = \boldsymbol{Y}^{(L)}$.

To calculate $\partial Z_i(\tau)/\partial\tau$, we introduce a notation

$$\boldsymbol{y}^{(j)} = \frac{\partial Z_i(\tau)}{\partial\boldsymbol{Y}^{(j)}}, \tag{16}$$

and employ a chain rule to obtain a backward operation for $\boldsymbol{y}^{(j)}$ as

$$\begin{aligned} \boldsymbol{y}^{(j-1)} &= \left(\frac{\partial\boldsymbol{Y}^{(j)}}{\partial\boldsymbol{Y}^{(j-1)}}\right)^{\top} \boldsymbol{y}^{(j)} \tag{17} \\ &= {W^{(j)}}^{\top}\text{diag}(f'^{(j)}(W^{(j)}\boldsymbol{Y}^{(j-1)} + \boldsymbol{b}^{(j)}))\boldsymbol{y}^{(j)} \qquad (j = 1, 2, \cdots, L), \tag{18} \end{aligned}$$

where we have $\boldsymbol{y}^{(L)} = 1$. By recursively applying the backward operation, we finally obtain the desired derivative as

$$\partial Z_i(\tau)/\partial\tau = [\boldsymbol{y}^{(0)}]_1, \tag{19}$$

where $[\boldsymbol{y}^{(0)}]_1$ represents the first element of the vector $\boldsymbol{y}^{(0)}$. We can then construct an extended network that produces both $Z_i(\tau)$ and $\partial Z_i(\tau)/\partial\tau$ by concatenating the original feedforwad network of eq. (15) and the backward network of eq. (18), as shown in Fig. S1.

### Evaluating the gradient of the loss function

The loss function of our model, the negative log-likelihood function, depends on both $Z_i(\tau)$ and $\partial Z_i(\tau)/\partial\tau$. The derivative $\partial Z_i(\tau)/\partial\tau$ is computed using backpropagation, as seen in the previous section. Then the computational graph to evaluate the loss function is then obtained as in Fig. S1. The gradient of the loss function with respect to the parameters can be evaluated by applying backpropagation to the computational graph.

Here we use backpropagation twice for training the model, the first for calculating $\partial Z_i(\tau)/\partial\tau$ and the second for calculating the gradient of the loss function. This kind of procedure is sometimes called as double backpropagaton [15].

Figure S1: A network structure of our model ($L = 4$). We omitted the computational graph for the operation of the RNN for simplicity.