[Reviews · NeurIPS 2019]

Reviewer 1



The paper proposes a neural network model for temporal point processes. The model uses a neural network rather than a parametric form to model conditional intensity function. To overcome the need to integrate the intensity function to compute likelihoods, the model estimates the cumulative intensity function, which can easily be differentiated. Experimental results on synthetic and real data sets show superior performance compared to state of the art. I believe this paper has the potential to be very impactful. The proposed model is clever and effective, yet simple to understand and implement.  My main reservation is that evaluation results were limited to likelihood-based metrics (log likelihood scores and intensity function curves). The paper would have been even stronger if it contained metrics that are tailored to concrete prediction tasks (e.g. time prediction error). Minor:L95: I understand that training with long sequences can be problematic due to gradient vanishing/explosion but the word "intractable" gives the impression that the problem is computational complexity. ====================================================== I thank the authors for their response and for conducting prediction experiments.

Reviewer 2



In general, I liked this approach. It is a new an interesting take on the problem and one that seems obvious in retrospect (which is often a sign of a good idea). I was happy to read the paper and feel that the idea should be generally communicated to the field as a whole. I am concerned that the paper fails to give the CT-LSTM model of [7] its full due, however. The introduction states that the hazard (or intensity) functions of previous work are either constant or have a rather fixed form (such as an exponential asymptote). [7] is a noteable exception to this. The hidden state does exponentially decay, but it is multi-valued (ie a vector) and a intensity is a non-linear function of the hidden state and therefore can have mroe complex behavior. While the introduction, as written, is true, it does not acknowledge this fact. Further, in Section 3 (Related works), this work is set to the side, stating that it performs very similarly to the RNN model of Du et al. This may or may not be true (my own experience is more mixed), but it is notable that the experiments do not compare to this single other method that could produce more complex intensity functions. If the authors had backed up such a statement (that CT-LSTM does not do as well) with experimental results showning it, this paper would be *significantly* stronger. As it is, I am left wondering if the other non-exponentially decaying method (CT-LSTM) would do as well as the proposed method in this paper. Secondly, I am concerned about the training procedure for the "exponential" model in the experimental results. This uses the intensity function of Equation 6. This can exactly model a single exponential kernel Hawkes process. Even for a HP with a kernel that is the mixture of two exponentials, the single exponential can often do reasonably well. Yet, these results are worse than for the piecewise constant model and Figure 3 suggests it has not fit the parameters of the exponential properly (although this is hard to tell, as this might be a testing example). That the exponential model *can* fit these models well does not, of course, state that in practice it will. However, if this comes down to a matter of the training/fitting procedure and whether one method is more robust than another, we need more detailed experimental results demonstrating this. As it is, I am left worried that the authors did not try "hard enough" to get the competing models to fit.

Reviewer 3



The proposed model mainly focuses on modeling the integration of intensity function. The paper is well written and easy to understand. The proposed model is mainly based on monotonic networks and technical novelty is a bit incremental as it is more like an application of monotonic networks to point processes. As for experimental evaluation, the paper only shows the improvement of log-likelihood evaluation, while a major application of point process models is prediction, such as event prediction and time prediction (see experiments in Du et al 2016). The intensity function is important in making this prediction, however it is not clear to me how to use this proposed network to derive the intensity function and make predictions, and this aspect is also not evaluated in the experiments. ----------------------------------------------------- the authors' response addresses my concerns, and I changed the initial score

[Author Response · NeurIPS 2019]

We greatly appreciate the three reviewers for their valuable comments. The following are our responses. We will add the following results and discussions in the final version of the manuscript.

**[RESPONSE TO REVIEWER #1]** (1) To evaluate the performances based on a metric other than the log-likelihood, we newly carried out the time prediction experiments; we predict the time $t_{i+1}$ of the next event from the times of the preceding events. We herein use the median $t_{i+1}^*$ of the predictive distribution to predict $t_{i+1}$ for each model. To obtain the median $t_{i+1}^*$, we use the relation $\Phi(t_{i+1}^* - t_i | \boldsymbol{h}_i) = \log(2)$, where $\Phi$ is a cumulative hazard function. This relation is derived from the property that the integral of the intensity function over $[t_i, t_{i+1}]$ follows the exponential distribution with mean 1. Then, the median $t_{i+1}^*$ can be efficiently obtained by solving the relation using a root finding method (e.g., bisection method); it takes only a second for our model to generate predictions for 20000 events. Therefore, the cumulative hazard function also plays a crucial role in generating a median predictor. The performance is evaluated by the mean absolute error, summarized below. In the table, the best score is in bold, and it is in red if the difference between the best score and the second best score is statistically significant ($p < 0.01$).

| | S-Poisson | N-Poisson | S-Renewal | N-Renewal | SC | Hakes1 | Hawkes2 | Finance | Call | Meme | Music | AVERAGE |
|---|---|---|---|---|---|---|---|---|---|---|---|---|
| constant | 0.696 | **0.704** | 1.085 | 0.450 | 0.543 | 0.915 | 1.075 | 0.883 | 0.966 | 0.848 | 1.066 | 0.839 |
| exponential | 0.696 | 0.716 | 0.917 | 0.452 | 0.498 | 0.854 | 0.986 | 0.840 | 0.922 | 0.823 | 0.788 | 0.772 |
| piecewise const. | **0.696** | 0.716 | 0.982 | 0.437 | **0.494** | 0.850 | 0.965 | **0.839** | 0.878 | 0.826 | 0.856 | 0.776 |
| neural network | 0.696 | 0.710 | **0.894** | **0.414** | 0.496 | **0.848** | **0.962** | 0.847 | **0.873** | **0.811** | **0.783** | **0.758** |

We find that our model performs better than the other models on average and that the performance of our model is best or close to the best for the most cases. These results demonstrate the effectiveness of our model in the prediction task.

**[RESPONSE TO REVIEWER #2]** (2) We recognized the importance of the CT-LSTM model [7] in the current context, and we will discuss more on it in the final version of the manuscript. We newly compared the performances between our model and the CT-LSTM model. The performance is evaluated in terms of the negative log-likelihood per event calculated for the test data. The following table lists the scores of the CT-LSTM model relative to our model; the positive score means that our model is better than CT-LSTM model. In the table, the score is in blue if the difference between the two models is statistically significant ($p < 0.01$).

| | S-Poisson | N-Poisson | S-Renewal | N-Renewal | SC | Hakes1 | hawkes2 | Finance | Call | Meme | Music | AVERAGE |
|---|---|---|---|---|---|---|---|---|---|---|---|---|
| CT-LSTM | -0.002 | -0.013 | 0.012 | -0.007 | -0.008 | -0.002 | 0.002 | -0.011 | 0.060 | 0.590 | 0.163 | 0.071 |

We find that our model performs much better than the CT-LSTM model for the Meme and Music datasets and that the performances are very close to each other for the other cases. Accordingly our model performs better than the CT-LSTM model on average. We also evaluated the predictive performances in terms of the mean absolute error. The average error over all the datasets is 0.766 for the CT-LSTM model and 0.758 for our model, and the difference is statistically significant ($p < 0.01$). These results indicate the superior performance of our model.

The technical advantage of our model is that the cumulative intensity function is directly modeled by a neural network, which makes the estimation and prediction efficient, and our model is also relatively easy to implement. For the CT-LSTM model, the intensity function is modeled as a nonlinear function of the time-evolving memory cells, and the estimation and prediction are simulation-based, which can be computationally expensive and can deteriorate the performance.

(3) The exponential model cannot fit to the Hawkes1 model perfectly because the hazard functions are slightly different from each other. While the hazard function of the exponential model is an exponential function, that of the Hawkes1 model is the sum of an exponential function and a constant $\mu$ (see line 211).

**[RESPONSE TO REVIEWER #3]** (4) A method to derive an intensity function is described in detail in the original submission (please see lines 134-140 in the manuscript and lines 3-20 in the supplementary text).

(5) In *RESPONSE TO REVIEWER #1*, we described an efficient method to make a median prediction based on the cumulative hazard function, and we additionally evaluated the predictive performances based on the mean absolute error, which finally showed the effectiveness of our model.

(6) The reviewer #3 commented that "*The proposed model is mainly based on monotonic networks and technical novelty is a bit incremental ...*". However, the novelty of our study is, we believe, rather that our approach enables us to model an intensity function in a flexible way and to exactly and efficiently compute the log-likelihood, which is important for application. The monotonic network is just a component of our new approach. This method is clearly advantageous against the studies that use a parametric hazard function or the studies [6, 7] that rely on numerical approximations to evaluate a model with a flexible intensity function, which can be computationally expensive and can deteriorate the performance. Our study therefore demonstrates the effectiveness of a neural network for the modeling of point processes. We hope that this point is positively evaluated.

[Meta-Review · NeurIPS 2019]

The reviewers appreciated the contribution and they think the work advances the state of the art and may spur interest in the temporal point process community within NeurIPS. I encourage the authors to take into account the reviewers' comments while preparing the final version, in particular, in regards to the experimental evaluation, where there is room for improvement.